# Estimation of free-roaming domestic dog population size: Investigation of three methods including an Unmanned Aerial Vehicle (UAV) based approach

Charlotte Warembourg[1]*, Monica Berger-González[2,3], Danilo Alvarez[2],
Filipe Maximiano Sousa[1], Alexis López Hernández[2], Pablo Roquel[2], Joe Eyerman[4],
Merlin Benner[5], Salome Dürr[1]

1 Veterinary Public Health Institute, Vetsuisse Faculty, University of Bern, Bern, Switzerland, 2 Universidad del Valle, Guatemala City, Guatemala, 3 Swiss Tropical and Public Health Institute, University of Basel, Basel, Switzerland, 4 RTI International, Seattle, Washington DC, United States of America, 5 Remote Intelligence, LLC, Wellsboro, Pennsylvania, United States of America

☉ These authors contributed equally to this work.
* charlotte.warembourg@vetsuisse.unibe.ch

**Data Availability Statement:** All data collected in Guatemala is bound to the data protection bylaws

## Abstract

Population size estimation is performed for several reasons including disease surveillance and control, for example to design adequate control strategies such as vaccination programs or to estimate a vaccination campaign coverage. In this study, we aimed at investigating the possibility of using Unmanned Aerial Vehicles (UAV) to estimate the size of free-roaming domestic dog (FRDD) populations and compare the results with two regularly used methods for population estimations: foot-patrol transect survey and the human: dog ratio estimation. Three studies sites of one square kilometer were selected in Petén department, Guatemala. A door-to-door survey was conducted in which all available dogs were marked with a collar and owner were interviewed. The day after, UAV flight were performed twice during two consecutive days per study site. The UAV's camera was set to regularly take pictures and cover the entire surface of the selected areas. Simultaneously to the UAV's flight, a foot-patrol transect survey was performed and the number of collared and non-collared dogs were recorded. Data collected during the interviews and the number of dogs counted during the foot-patrol transects informed a capture-recapture (CR) model fit into a Bayesian inferential framework to estimate the dog population size, which was found to be 78, 259, and 413 in the three study sites. The difference of the CR model estimates compared to previously available dog census count (110 and 289) can be explained by the fact that the study population addressed by the different methods differs. The human: dog ratio covered the same study population as the dog census and tended to underestimate the FRDD population size (97 and 161). Under the conditions within this study, the total number of dogs identified on the UAV pictures was 11, 96, and 71 for the three regions (compared to the total number of dogs counted during the foot-patrol transects of 112, 354 and 211). In addition, the quality of the UAV pictures was not sufficient to assess the presence of a mark on the spotted dogs. Therefore, no CR model could be implemented to estimate the size of the

and confidentiality agreements requested by the Ethics Committee of the Center for Health Studies (CES-UVG) of the Universidad del Valle de Guatemala. Given special permission had to be requested to Municipal authorities in Poptún and to two indigenous Maya Q'eqchi' Councils due to potentially sensitive information, access to further data can be granted upon request following strict guidelines. For such a request please contact the CES-UVG Ethics Committee president, Dr. Jorge Jara at jjara@ces.uvg.edu.gt or phone +502 2364 0492.

**Funding:** Funding for this research was provided by the Albert-Heim-Stiftung Project Nr. 132 – (http://www.albert-heim-stiftung.ch, awarded to SD), the Spezialisierungskommission of Bern University (SpezKo) (https://www.vetsuisse.unibe. ch/weiterbildung/spezialisierungskommission/ index_ger.html, awarded to FMS), the Wolfermann-Nägeli-Stiftung Nr. 2018/28 (awarded to SD), the Swiss Programme for Research on Global Issues for Development (r4d programme), Project Nr. 160919, (http://p3.snf.ch/project-160919) and Vontobel Stiftung, grant 27.04.2017 (https://www.vontobel.com/en-ch/wealth-management/our-solutions/additional-services-and-themes/charitable-foundations - awarded to MBG). The two UAV used during the study were provided by RTI International and the Universidad del Valle. RTI International provided support in the form of salaries for author MB, but did not have any additional role in the study design, data collection and analysis, decision to publish, or preparation of the manuscript. The specific roles of this author is articulated in the 'author contributions' section.

**Competing interests:** Merlin Benner is an owner and employee of Remote Intelligence, LLC, a private, forprofit company, which provide Unmanned Aerial Drone sales and consulting services. This commercial affiliation does not alter our adherence to all PLOS ONE policies on sharing data and materials

FRDD using UAV. We discussed ways for improving the use of UAV for this purpose, such as flying at a lower altitude in study area wisely chosen. We also suggest to investigate the possibility of using infrared camera and automatic detection of the dogs to increase visibility of the dogs in the pictures and limit workload of finding them. Finally, we discuss the need of using models, such as spatial capture-recapture models to obtain reliable estimates of the FRDD population. This publication may provide helpful directions to design dog population size estimation methods using UAV.

## Introduction

Estimating the size of domestic and wild animal populations has been used in diverse research fields, such as for species monitoring and conservation [1,2] or disease surveillance and control [3]. For example, population size estimation methods have been used to monitor biological diversity in space and time [4], estimate the abundance and conservation status of rare and endangered species such as whales or rare and elusive carnivores [5,6] or to design and conduct vaccination programs [7,8] and estimate vaccination campaign coverage for infectious diseases [9].

Methods have been developed to estimate the size of both wild [10] and domestic animal populations [11,12]. Among domestic animals, stray dog populations have been widely studied because of their importance in the transmission of rabies, a deadly zoonotic disease [13–17]. According to the World Health Organization (WHO), more than 99% of human rabies cases worldwide are caused by dog bites [18]. The World Organization for Animal Health (OIE) defines stray dogs as dogs not under direct control by humans or not prevented from roaming freely. They are divided into three categories: free-roaming owned dogs, free-roaming owner-less dogs still living within human communities and feral dogs which have reverted to a wild state and do not depend on humans anymore [19]. The dogs of the first two categories are also named as "free-roaming" or "free-ranging", which encompass both owned and ownerless dogs [20] and are more relevant for zoonotic disease control than feral dogs due to their closeness to the human population.

The estimation of free-roaming domestic dog (FRDD) population size is challenging and susceptible to bias because of the presence of heterogeneity in the dog population (i.e. presence of owned and ownerless dogs) which leads to heterogeneity in detection probabilities [13]. The most commonly used methods are based on census surveys [21,22], distance sampling (transect based method where perpendicular distances from random transect lines to the dogs are measured to estimate the dog density) [14,23] and capture-recapture (CR) technics [14–17,20,24].

Guatemala is a Central American country where canine rabies is still endemic. Ministry of Health reported 956 animal rabies cases and 18 human rabies deaths in the country during the period of 2005–2017 [25]. The current dog population size in Guatemala is unknown. A study conducted in 2008, focusing on owned dogs, estimated the number of dogs in 12 neighborhoods of Todos Santos Cuchumatán, a town located in the Department of Huehuetenango, at 392 [26]. The method chosen was a door-to-door household census. Based on a study published in 1959 [27], the animal health authorities in Guatemala currently use a human: dog ratio of 5:1 to estimate the number of vaccines needed for the national rabies vaccination campaigns. However, the human: dog ratio may vary regionally, which was also found as preliminary outcome of currently ongoing studies undertaken by the Universidad del Valle (UVG)

that aim to estimate the human: dog ratio in urban, peri-urban and rural communities (unpublished data). The methods used so far to estimate the dog population size in Guatemala focused on owned dogs only and did not take into account the potential presence of ownerless dogs within the population.

Thanks to recent progress in technology and software development, the utilization of Unmanned Aerial Vehicles (UAV) that can take geo-referenced pictures and automatically fly along pre-defined paths has strongly increased for population monitoring [28,29]. Commonly called drones, they became a valuable tool in agricultural, environmental and wildlife monitoring [30]. Some studies used UAV to estimate population sizes of various animal species including sea birds [28,31], marine fauna [28,32], biungulates (red deer, roe deer, elk, wild boars, bison) [33], wolves [34] and marsupials [30]. Comparing to other counting methods, UAV's advantage is the ability to count wildlife time efficiently in remote areas and in a standardized and reproducible way [34,35]. These features may be of interest for FRDD counting, especially for dogs that do not have an owner and might have an elusive behavior with human beings. The availability of inexpensive UAV is increasing [30]. Studies in ecology suggested that, compared to traditional methods, the utilization of UAV might improve the data collection efficiency and cost-effectiveness [28]. The utilization of affordable UAV could lower the cost of data collection compared to traditional ground survey while providing accurate and reproducible results. To the best of our knowledge, this is the first study published that evaluates the possibility of using UAV to estimate the size of FRDD populations.

The aim of this study is to investigate three methods used to estimate the size of stray dog populations in three locations in Guatemala. These are a method based on UAV transect survey, a method based on foot-patrol transect survey fit into a CR model and the utilization of the human: dog ratio. Practicability and ethical considerations of the use of drones will be discussed as well.

## Material and methods

### Study area

The study was performed in the Petén Department of Guatemala, located in the Northern lowlands of the country (Fig 1), specifically in the Poptún Municipality, one of 14 political subregions. It covers approximately 69,000 square kilometers (31.8% of Guatemala's surface) and constitutes the southern part of the Yucatan tableland [36]. It is a sub-tropical forest biome with thick carbonate deposits responsible for a predominant karst topography, creating thin soils that erode easily and are not fit for agriculture [37]. Once scarcely populated, this region has seen mass migration patterns increase in the last 30 years, causing an accelerated land-use change associated to large scale African palm production, cattle industry and slash-and-burn agriculture [38]. Population in the region is 69% Ladino (Spanish-speaking) and 30% Maya, with an overall illiteracy rate of 22%. Close to half of the population is living under the poverty line [39]. Most of the populated areas are considered rural (59.46%), where access to basic services such as safe drinking water, electricity, sanitation, public health facilities, or transportation is limited [39].

Three study areas were selected within the Poptún Municipality, depending on their expected dog population sizes. Overall, Poptún has 52,282 inhabitants according to the 2018 census (INE, 2019, data accessed on Oct 10, 2019). The main city of Poptún, categorized as an urban area (Fig 2), has an estimated population of 29,000 people. Sabaneta and La Romana are two villages north of Poptún with 804 and 485 residents, respectively, surrounded by agricultural landscape (rural areas) (Fig 2). Sabaneta and La Romana were selected by convenience as they were already involved in a research project named "One Health Poptún" (r4d grant

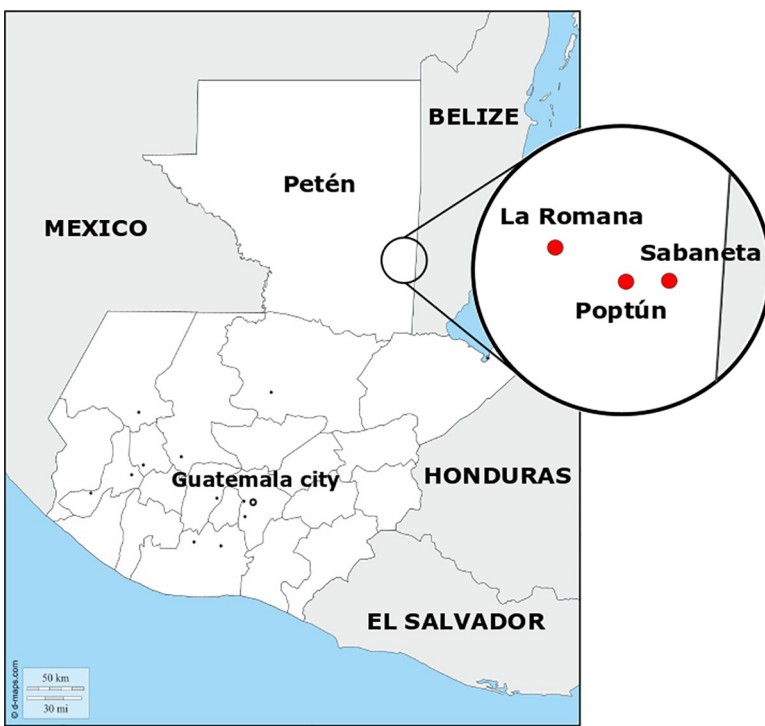

**Fig 1. Localization of the three study areas (Poptún, Sabaneta and La Romana) in Petén department, Guatemala.**
Source: d-maps.com.

SwissTPH-UVG) implemented by researchers of the UVG (https://www.ces.uvg.edu.gt/page/2018/02/07/proyecto-una-salud-poptun-video/). This project aimed to improve the health situation of Maya communities in the Poptún Municipality using a One Health approach. Poptún city was chosen as it was the closest urban area to Sabaneta and La Romana.

As part of "One Health Poptún" research project, a human and domestic animal census was performed to serve as a baseline for a surveillance and response program in Sabaneta and La Romana. The dog census baseline study was conducted during June and July 2017 and consisted in asking the number of dogs owned by each participating household. At the beginning of the study, around 20% of the households refused to participate. Some of them were included in the study at a later stage, until April 2018 in Sabaneta and July 2018 in La Romana. The

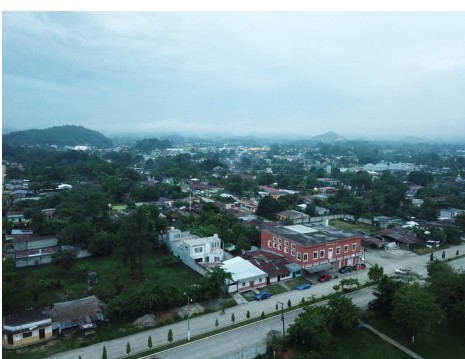 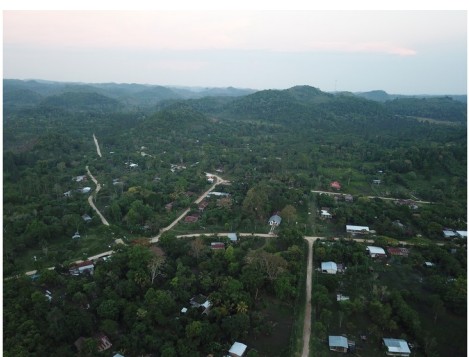 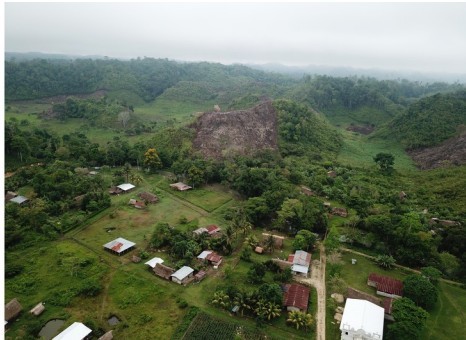

**Fig 2. Aerial pictures of the three study areas taken by a Mavic Pro in May-June 2018.** Left: Poptún, Center: Sabaneta, Right: La Romana.

number of dogs living in those households, at the time of their inclusion in the study was recorded and added to the baseline data. In total, 99% (118/119 and 98% (197/201) of the households accepted to participate in the study and 110 and 289 owned dogs were recorded, in La Romana and Sabaneta respectively (unpublished data). No dog census data was performed in Poptún by the project.

For each of the three study areas, we defined a one square kilometer study site, in which the dog population estimation was performed. In La Romana and Sabaneta, the study sites covered the entire village; in Poptún, it only covered part of the city. The GPS coordinates of the three study sites are available in S1 Table. As study population, we considered all type of dogs visible in the streets or yards regardless of their ownership status. It therefore constituted of owned or ownerless FRDD or stray dogs as described in the OIE guidelines on stray dog population [19].

The study was conducted between May and June 2018.

## Ethical approval

Given the study design included research with human subjects as well as with animals, the research team submitted two ethical protocols prior to conducting fieldwork, one to UVG's International Animal Care and Use Committee (IACUC) (Protocol No. I-2018(3)), and another to the Ethics Review Board (ERB) of the Committee for Research on Human Subjects of the Center for Health Studies in UVG (Protocol No. 175-04-2018). Since the methodology included the use of UAV to fly over both private and public areas in each town, it was necessary to obtain permits from the National Authority for Civil Aeronautics (DGAC in Spanish), which included registering each drone with a unique identification number. Given research with UAVs is relatively new in Guatemala, the DGAC had no written regulations pertaining to ethical considerations for their use in contexts like the one addressed by this project, a void commonly reported in the scarce literature on UAVs for research [40]. The lack of clear guidelines to address issues of privacy, data protection and public safety, along with conducting research in indigenous populations considered vulnerable [41], demanded this project followed consuetudinary law as practiced in each site. The research design was presented and discussed with health and political authorities of Poptún in several meetings until all concerns for public safety were addressed. Official research permits were then requested and obtained from the Regional Health Authority (Petén Sur Oriente) of the Guatemala Ministry of Health and from the Municipal Government of Poptún. The latter granted specific permits to conduct research in the urban area and directed the communication campaign to inform citizens about the UAV flights, selection of homes and related activities. While the urban area is ruled only by municipal law and has a predominant Ladino population, the towns of Sabaneta and La Romana are predominantly Maya Q'eqchi' (65% and 98% respectively) and are ruled by traditional indigenous consuetudinary law. In both sites, the team met with local leaders from the Community Development Councils (COCODES) during March and April 2018 to explain the aims and methods of the research, address concerns, and request formal authorization for implementation. In La Romana, the COCODE asked that the team returns on a following weekend to explain to the entire population the research endeavor in an Open Council, in the local language, where a voting and consensus process was conducted resulting in the community's approval of the project. This decision was formally communicated to the research team one week later in writing by the COCODE leaders. In Sabaneta, the COCODE authorities determined the intervention was of value to the community, but requested a door-to-door campaign to explain to each family head the proposed research. Once this process was concluded, the COCODE extended a formal letter of approval to conduct the research. The

process was led by anthropologists and supported by Maya Q'eqchi' staff of UVG to guarantee cultural pertinence, enhance two-way communication, build trust, and understand compliance to consuetudinary law. Local community authorities were involved in on-site supervision of all research activities.

## Dog population size estimation

Three methods were used to estimate the dog population sizes in the three study areas: a method based on a UAV transect survey, a method based on a foot-patrol transect survey and a method based on the utilization of the human: dog ratio. Because the two first methods cannot detect the entire dog population (e.g. dogs located under trees or roof overhangs for the UAV and dogs located too far away from the transect lines for the foot-patrol survey), the data aimed at being analyzed using CR models accounting for the local dog ecology. Before the UAV flights and foot-patrol transect survey, a marking and an interview survey were performed. The interview survey aimed at collecting information on some aspects of the dog ecology, including dog keeping practices and ownership status of dogs in these areas.

**Marking and interview survey.** All households located within the three study sites were visited and dog-owning households were identified. In each dog-owning household, we interviewed the dog owner and aimed to mark all dogs of the household by a collar. A written consent form was obtained from each dog owner. Dog handling was performed by or under the supervision of a trained veterinarian, and all efforts were made to minimize stress. The collars used for marking the dogs were red and carried a grey box containing a GPS chip (used for another study, no meaning here), easily visible from several dozen meters. In Sabaneta and La Romana, we collared both free-roaming dogs and confined dogs if they could be in contact with FRDD dogs (e.g. restrained by a chain in an open yard). In Poptún, due to the high number of dogs, the team focused the collaring on free-roaming dogs and not on fully confined dogs. The reasons for non-collaring of the dogs were refusal of the owner to participate to the study, dog's stress or aggressiveness, or absence of the owner or the dog during the visit. The main reason being the absence of the owner or the dog during the visit. Only a small proportion of the households refused to participate. Most of the households that refused to participate in this study had previously refused to be part of the "One Health Poptún" project. In most cases, it was the belief that a) drawing blood would get people and dog sick, b) we would inject them with the "mark of the devil" (belief related to the interpretation of a quote from the Bible) and c) had prior experiences with people injecting harmful products that lead to the death or sickness of their pets. Blood samples were taken from humans and animals as part of "One Health Poptún" and from dogs in this study (used for another study, no meaning here). In addition, one person thought that the collar would disturb the dogs while hunting. Dogs younger than four months, pregnant bitches and sick animals were excluded from the study and not marked. The reasons for this was the too small size of the puppies for the collars and to avoid unnecessary stress of pregnant or sick animals. Information on age, potential dog pregnancy and health status of the dog were asked to the owner before the marking but not formally recorded.

The interviews were conducted using a questionnaire collecting information on dog confinement practices and owner estimates on the number of owned dog, community dog and ownerless dog populations in their living area (S2 Table). A dog with a dedicated owner is considered as an owned dog, a dog without owner but fed by the community is considered as a community dog and a dog without owner and not fed by the community or a particular person is considered as an ownerless dog. The data were directly put in a digital format into a tablet equipped with KoBoCollect Android application (https://www.kobotoolbox.org) and later

downloaded as an Excel spreadsheet. In total, 61 dogs were marked in La Romana, 125 in Sabaneta and 117 in Poptún study site, within 39, 76 and 73 households respectively, which corresponds to the number of interviews conducted. The data collection lasted for four days in Poptún and Sabaneta study sites and three days in La Romana.

**Method 1: UAV transect survey.** The UAV used in this study was a DJI Mavic Pro (https://www.dji.com/ch/mavic). The main characteristics of its aircraft and camera are described in Table 1.

A total of twelve flights were scheduled, four per study site. In each site, the flights were conducted twice a day, in the morning and late afternoon, during two consecutive days following the marking. It has been suggested in previous studies that free-roaming dogs have peaks in activity early morning after sunrise and late afternoon before sunset [42]. The flight course was mapped in advance using Pix4Dcapture, a free mobile application for drone flight planning (https://www.pix4d.com/product/pix4dcapture). A grid with parallel transect lines covering the predefined flight area were defined automatically by the application. The number of transect lines was conditioned by the altitude of the flight and pictures' overlap selected by the operator. Accordingly, the number of pictures is given by the altitude of the flight and pictures' overlap, ensuring that the entire study site surface is captured by the pictures. Those parameters were dependent on the number and capacity of batteries available. Five batteries (capacity: 3038 mAh, voltage: 11.4V, battery type: LiPo 3S, Net weight: approx. 240g) were available per flight.

**Table 1. Technical specifies of the Mavic Pro used for the UAV flights.**

| **AICRAFT** | |
| --- | --- |
| Size folded | H83mm x W83mm x L198mm |
| Diagonal Size (Propellers Excluded) | 335 mm |
| Weight (Battery & Propellers Included) | 1.62 lbs (734 g) (exclude gimbal cover) 1.64 lbs (743 g) (include gimbal cover) |
| Max Ascent Speed | 16.4 ft/s (5 m/s) in Sport mode |
| Max Descent Speed | 9.8 ft/s (3 m/s) |
| Max Speed | 40 mph (65 kph) in Sport mode without wind |
| Max Service Ceiling Above Sea Level | 16404 feet (5000 m) |
| Max Flight Time | 27 minutes (no wind at a consistent 15.5 mph (25 kph)) |
| Max Hovering Time | 24 minutes (no wind) |
| Overall Flight Time | 21 minutes (In normal flight, 15% remaining battery level) |
| Max Total Travel Distance (One Full Battery, No Wind) | 8 mi (13 km, no wind) |
| Operating Temperature Range | 32˚ to 104˚ F (0˚ to 40˚ C) |
| Satellite Positioning Systems | GPS / GLONASS |
| **CAMERA** | |
| Sensor | 1/2.3" (CMOS), Effective pixels:12.35 M (Total pixels:12.71M) |
| Lens | FOV 78.8˚ 26 mm (35 mm format equivalent) f/2.2 Distortion < 1.5% Focus from 0.5 m to ∞ |
| ISO Range | photo: 100–1600 |
| Electronic Shutter Speed | 8s -1/8000 s |
| Image Size | 4000×3000 |
| Supported File Systems | FAT32 (≤ 32 GB); exFAT (> 32 GB) |
| Format | JPEG, DNG |
| Supported SD Cards | Micro SD™ Max capacity: 128 GB. Class 10 or UHS-1 rating required |
| Operating Temperature Range | 32˚ to 104˚ F (0˚ to 40˚ C) |

Table 2 shows the overlap, altitude and number of pictures taken per flight in the three study areas.

The pictures were automatically stored in the UAV's SD card.

In each study site, a team of two persons conducted the UAV flights, the operator and the observer. The operator was in charge of the controller and monitored the flight on the control screen. The camera view was displayed on the control screen and a signal was transmitted each time a picture was taken. The operator checked that the transmission between the UAV and the controller was functioning well and pictures were regularly taken. There is no need to drive the UAV as it flies automatically following the flight course defined on Pix4Dcapture application. The observer was standing next to the operator. He was visually controlling that the UAV was regularly visible, operating well and did not enter in collision with a bird, a tree or an antenna. When the UAV ran out of battery, it automatically returned to the take-off spot and the observer replaced the battery. Then, the UAV flew back to the previous location and pursued with the data collection. The mean time for conducting the transect survey was 42 minutes in La Romana, 46 minutes in Sabaneta and 63 minutes in Poptún study area.

The analysis of the pictures (Fig 3) was undertaken after the data collection period by researchers from the UVG Arbovirus and Zoonosis group. The first step was to eliminate pictures that could not be analysed due to poor quality (blurred or taken at too high altitude) or fully covered by forest canopy (Fig 4). This exclusion phase was done by three people simultaneously. The rest of the pictures was analysed individually to identify dogs. The pictures to be further reviewed were then stored electronically sorted by study area, date and time.

To identify dogs including their location, each picture was divided into nine (3 x 3) quadrants and each quadrant was identified with a combination of letters and numbers (Fig 5). Each picture and quadrant was reviewed in a zigzag pattern, from top left to bottom right, and identified dogs were marked with a red circle. When there was any doubt about the identification of a dog in any of the pictures, these were analysed again by three different people to have a consensus on whether the observation was a dog or not. Pictures of identified dogs and doubtful observations are shown on Fig 6. Doubtful observations resembling dogs but not presenting dog characteristics, such as ears, thorax, pelvis, limbs or tails, were not counted as dogs. This criteria was used as a guidance for the reviewers to identify dogs. Observations resembling dogs but presenting only part of the dog characteristics cited above were counted as dogs.

The study area, identification number of the picture, date and time when the picture was taken, identification of the dog, location of the dog in the quadrant, general observations on the dog (e.g. fur colour, presence of a collar), reviewer's name and date of review were entered in an Excel document for each identified dog (S3 Table).

**Method 2: Foot-patrol transect survey.** Simultaneously to the UAV flights, a second team performed foot-patrol transects to count dogs in the streets and opened yards where dogs had access to the street. The team was constituted of at least three persons: a local guide, a person recording the number of dogs and a person spotting the dogs. The transect lines were defined in advance on Google Earth according to the following rules (S1 Fig). The minimum distance between two lines was 100 meters, a transect line should not cross another line and we respected a buffer zone at the edge of the site of 100 to 400 meters to avoid counting dogs whose owner is not living in the study site. While counting the dogs, we distinguished between marked (collared dogs spotted during the transect walks) and non-marked dogs (dogs spotted during the transect walks not carrying the study collars). To avoid double counting of individual dogs, we took pictures or videos of the dogs encountered (Fig 7). During the transects, if the team had a doubt whether a dog was being seen for the second time, they would go through the pictures taken previously, check if the dog had already been counted and decide whether

**Table 2. Characteristics of the UAV flights performed in Petén department, Guatemala in May-June 2018.** Front and side overlap are defined by the overlap area of two consecutive pictures along the flight line (front) and between transect lines (side). In Poptún, only three flight could be realized because of bad weather conditions during the last flight. The difference of flight time between sessions having the same settings could explained by differences in the light condition. The system automatically adjusts flight speed depending on ambient light conditions to avoid motion blur.

| | Starting time | Ending time | Altitude (m) | Front overlap (%) | Side overlap (%) | Number of pictures |
|---|---|---|---|---|---|---|
| La Romana | 8:59 | 9:25 | 100 | 20 | 20 | 154 |
| | 17:15 | 17:41 | 100 | 20 | 20 | 154 |
| | 7:37 | 8:34 | 100 | 30 | 30 | 174 |
| | 17:20 | 18:20 | 100 | 35 | 35 | 181 |
| Sabaneta | 7:31 | 8:27 | 55 | 30 | 30 | 540 |
| | 18:05 | 18:36 | 55 | 30 | 30 | 318 |
| | 6:26 | 7:13 | 55 | 30 | 30 | 349 |
| | 17:12 | 18:02 | 55 | 30 | 30 | 378 |
| Poptún study site | 7:06 | 7:58 | 60 | 30 | 30 | 475 |
| | 16:46 | 18:10 | 60 | 30 | 30 | 425 |
| | 16:49 | 17:40 | 60 | 30 | 30 | 450 |

they count or ignore it. The total transect length was 3 kilometers in La Romana, 4.6 kilometers in Sabaneta and 8.2 kilometer in Poptún study sites and the mean time for conducting the transect was 58 minutes, 71 minutes and 128 minutes, respectively.

A CR model (S1 File) was developed to estimate the dog population size based on the number or marked and unmarked dogs spotted during the foot-patrol transects. This model was adapted from the model developed by Kayali et al [24] and used in a modified version by Gsell et al [15]. The data required are the number of dogs marked during the household study,

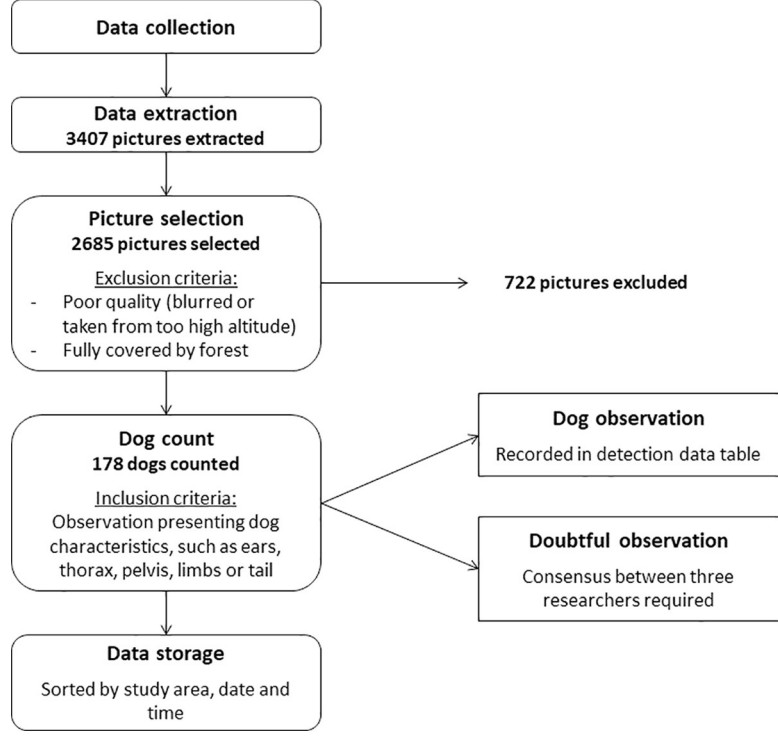

**Fig 3. Flow chart of the UAV pictures analysis process.**

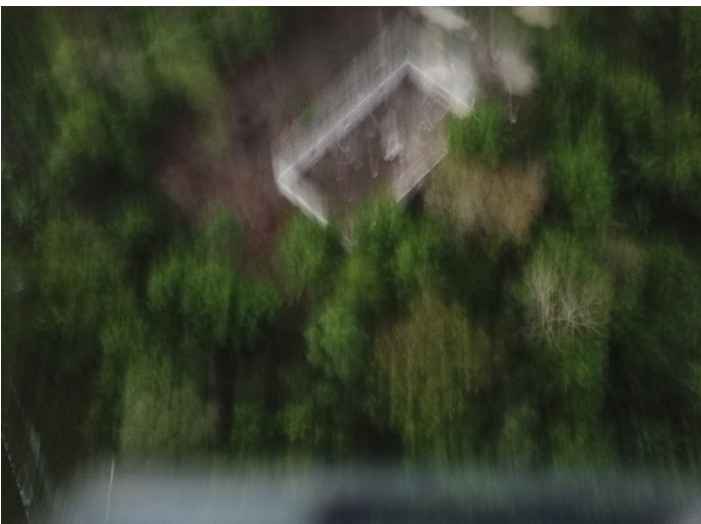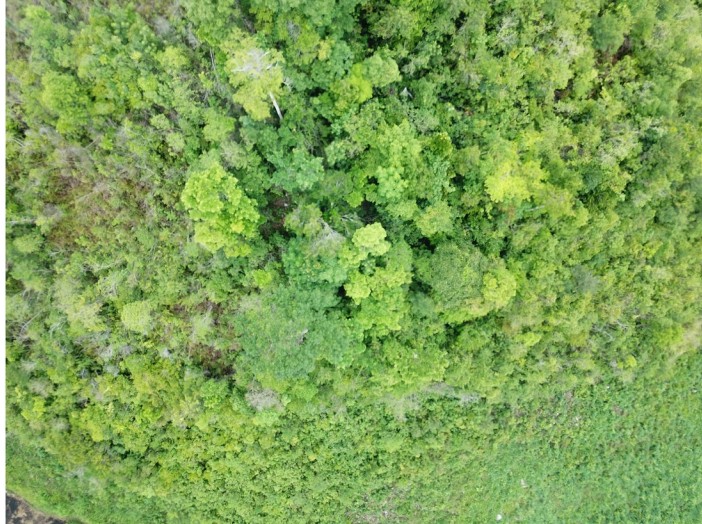

**Fig 4. Example of pictures excluded from the UAV picture analysis.** Left: blurred picture, Right: picture fully covered by forest canopy.

number of dogs spotted during the transect walks (distinguishing between marked and unmarked dogs), the number of collars lost during the study, the recapture probability and study site specific prior information derived from the questionnaire data (i.e. dog keeping practices and information about the ownerless dog population). The model considers the recapture probability, the probability for a marked dog to loose the marking (i.e. collar lost by the dog or removed by a person), the probability of a dog being confined (and therefore not seen) at the time of the transect walks and the probability of encountering ownerless dogs during the transects. It was fit into an Bayesian inferential framework.

All the marked dogs encountered during the transect walks were owned, whereas the unmarked dogs either were owned (but unmarked) or ownerless. We considered that the number of marked dogs, owned unmarked dogs and ownerless unmarked dogs counted during the transects follow a binomial distribution [24].

The probability of recapture was defined as a uniform distribution and combines the foot-patrol survey coverage, the encountering and the recording probabilities [15,24]. A minimum and maximum value were considered for each of the three factors and combined to obtain the minimum and the maximum value of the recapture probability uniform distribution (Table 3). To estimate the minimum coverage, we assume that 25m on each side of the transect lines were covered (i.e. dogs being within this distance will be observed). This included the

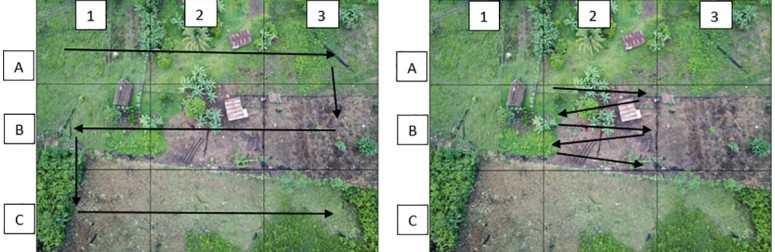

**Fig 5. Reviewing procedure of picture taken by the UAV in one of the rural sites.** The left picture shows the nine quadrants and the zigzag pattern used to review the picture. The right picture shows the zigzag pattern used to review each quadrant.

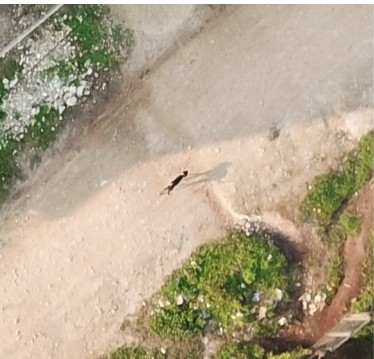
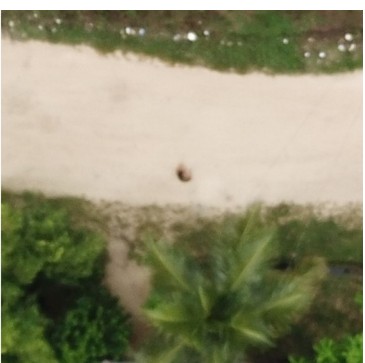

**Fig 6. Example of identifiable dogs and doubtful observations.** Left: dog, Right: doubtful observation.

road, as well as the yards of the households on each side of the road [24]. Since the transect lines covered almost every parallel road, we considered for the maximum value that the observers could see all the dogs between two transects lines. The size of the minimum and maximum coverage area were quantified using QGIS. The encountering and recording were taken from previous studies using the similar model by Kayali et al, and Dürr et al in Chad [24,43] and by Gsell et al in Tanzania [15]. This is based on observations that many dogs gather around their households or use the streets to move around the village and could therefore be seen and that the recording probability was very high with at least three person walking the foot-patrol transects together [24].

The probability lm(t) that a marked dog lost its collar at transect number t was modelled as:

$$lm(t) = \frac{Ml}{Mc} x \frac{t}{T}$$

with Ml denoting the total number of collars lost during the entire study period (three in La Romana, two in Sabaneta, four in Poptún study site); Mc denoting the total number of dogs collared in the study (61 in La Romana, 125 in Sabaneta and 118 in Poptún study site) and T denoting the total number of transects per study site (four each in La Romana, Sabaneta and Poptún).

We considered that ownerless dogs have a confinement probability of 0. Prior data on the confinement probability of owned dogs were calculated based in the information collected during the interview survey (S4 Table). The dog owner was asked to explain when their dog usually roams. The answer was then categorized into "never free-roaming", "always free-roaming", "free-roaming during the day only", "free-roaming during the night only", "free-roaming a few hours per day only" (S5 Table). We considered confinement probabilities during the time of the transects of 1 for dogs of the category "never free-roaming", 0 for dogs of the category "always free-roaming", 0–0.25 (uniform distribution) for dogs of the category "free-roaming during the day only", 0.5 for dogs of the category "free-roaming a few hours per day", and 0.75–1 (uniform distribution) for dogs of the category "free roaming during the night only". The minimum and maximum confinement probability were calculated for each study area based on these probabilities, considering the lower and upper limit of the range, respectively, weighted by the number of dogs per category. The prior information of the dog confinement probability was included into the CR model as a study site specific uniform distribution ranging from the minimum to the maximum confinement probability (Table 3).

Prior data on the probability of encountering an ownerless dog during the transects was calculated based on the owners' estimates on the number of owned and ownerless dogs living in

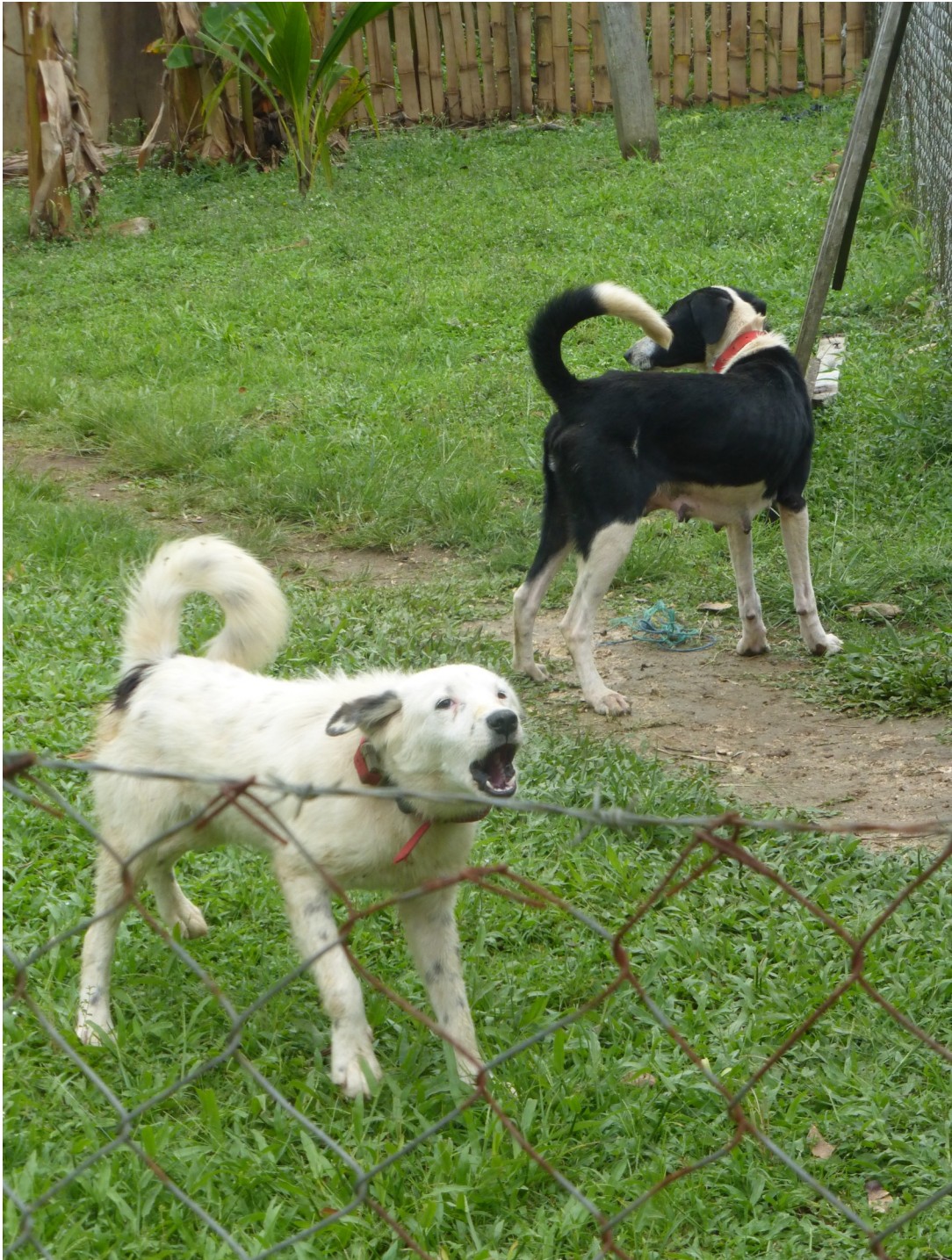

**Fig 7. Picture taken during the foot-patrol transect walks where marked dogs are visible.**

the study sites requested during the interview survey. We then calculated, for each study area, the ratio of ownerless to owned dogs using the estimation given by the respondents (S2 Fig). Log normal distributions were fitted on the data and distribution parameters were used as prior information incorporated in the statistical model (Table 3).

**Table 3. Prior information used in the capture-recapture model.**

| Parameter | | La Romana | Sabaneta | Poptún |
|---|---|---|---|---|
| Recapture probability (uniform distribution, min-max) | | 0.28–0.89 | 0.14–0.71 | 0.17–0.68 |
| | Coverage (min-max) | 0.44–1.00 | 0.23–0.80 | 0.26–0.76 |
| | Encountering (min-max) | 0.70–0.90 | 0.70–0.90 | 0.70–0.90 |
| | Recording (min-max) | 0.90–0.99 | 0.90–0.99 | 0.90–0.99 |
| Confinement probability (uniform distribution) | | | | |
| | Min | 0.133 | 0.157 | 0.293 |
| | Max | 0.142 | 0.157 | 0.320 |
| Ownerless to owned dog ratio (log normal distribution) | | | | |
| | μ | -6.57 | -7.71 | -7.67 |
| | T | 5.24 | 4.97 | 4.98 |
| Number of owned dogs living in the study site (uniform distribution) | | | | |
| | Min | 61 | 125 | 118 |
| | Max | 1000 | 1000 | 1000 |

No prior data was collected on the number of owned dogs during the interviews. Therefore, we intended to use non-informative priors. The number of owned dogs was modeled as a uniform distribution. The maximum value was set up at 1,000 in each study area, which was considerably higher than any expectations. A theoretical minimum value was considered as the number of dogs collared during the study in each study area (Table 3).

The model was implemented in OpenBugs (http://www.openbugs.net) version 3.2.3. The code of the model is presented in S1 File.

**Method 3: The human: Dog ratio.** Applying the human: dog ratio on the human population size in each study area builds the third method to estimate the dog population size. We used the human: dog ratio currently used by the health authorities in Guatemala for the annual national rabies vaccination campaign, 5:1 [27].The number of humans was derived from the total human census performed by the project "One Health Poptún" in Sabaneta and La Romana, leading to a population size of 804 and 485 people, respectively. The human census was performed as the same time than the dog census survey. Each participating household (99% in La Romana and 98% in Sabaneta) was asking about the member of people living in the household. Because of lack of human population data in Poptún study site, we could not estimate the size of the dog population in Poptún study area using the human: dog ratio method.

## Results

Out of the twelve flights, only eleven could be realized because of bad weather conditions (rain) during the third UAV flight in Poptún. A total of 3,407 pictures were reviewed from the

**Table 4. Number of dogs counted during UAV flights in comparison with the number of dogs spotted during the ground transect walks in three study areas of Poptún Municipality, Petén department, Guatemala in May-June 2018.**

| Transect number | La Romana | | Sabaneta | | Poptún study site | |
|---|---|---|---|---|---|---|
| | UAV flight | Ground transects | UAV flight | Ground transects | UAV flight | Ground transects |
| 1st transect | 0 | 26 | 31 | 94 | 33 | 55 |
| 2nd transect | 0 | 29 | 2 | 80 | 13 | 69 |
| 3rd transect | 3 | 33 | 35 | 88 | - | 51 |
| 4th transect | 8 | 24 | 28 | 92 | 25 | 36 |
| Total | 11 | 112 | 96 | 354 | 71 | 211 |

11 flights: 520 in La Romana, 1,537 in Sabaneta, and 1,350 in Poptún study site. Out of them, 2,685 were selected for review, 293 from La Romana, 1,048 from Sabaneta and 1,344 from Poptún. It is not possible to differentiate confined and free-roaming dogs on the UAV pictures. Because of the poor quality of the UAV pictures, it was not possible to assess if a dog was marked (i.e. having a red collar) or not. The number of dogs countered per flight is shown in Table 4. The numbers of dogs encountered during the foot-patrol transect survey are also presented for comparison. Substantially more dogs were observed during the foot-patrol transect walks than on the pictures taken by the UAV.

Because of their small size and their tendency to stay at home before weaning, young puppies were not spotted on the UAV's pictures. When dogs were located in yards, it was not possible to differentiate free-roaming dogs and confined dogs.

The percentage of dogs spotted in each type of land cover is not proportional to the land coverage (Table 5). Only one dog in La Romana was spotted outside of roads and yards. If adjusted for land coverage, a high percentage of dogs were spotted on roads in Sabaneta and Poptún study area. For example in Sabaneta, 36% of the dogs were spotted on the roads while the roads only cover 5% of the study zone.

The CR model based on the foot-patrol transect survey estimated a total dog population size of 78 in La Romana, 259 in Sabaneta and 413 in Poptún study sites. The credibility intervals and the estimated number of owned and ownerless dogs are shown in Table 5. The number of marked and unmarked dogs recaptured during the transect walks is shown in S6 Table.

Using the human: dog ratio, we obtained an estimation of 97 dogs in La Romana and 161 dogs in Sabaneta (Table 6). These numbers were compared to the census data of owned dogs collected during the "One Health Poptún" project.

## Discussion

In this study, three methods estimating dog population size in three study areas in Guatemala were investigated. This is the first published study that applied UAV to collect data for population size estimation for this species. The number of dogs detected with the UAV was substantially smaller than with the foot-patrol transects. Unfortunately, due to the quality of the UAV pictures, it was not possible to differentiate marked and unmarked dogs, which prevented the utilization of a CR model to estimate the size of the FRDD population based on the UAV transects counts.

To correctly interpret the comparison of the results of the methods, we need to consider the heterogeneity of the dog population and which subpopulation each method is quantifying. The total dog population can be divided into owned and ownerless dogs (Fig 8). Further, the owned dogs are free-roaming (partly or completely), completely confined and visible (e.g. in a yard fenced with a grid) or completely confined and not visible at any time (e.g. under a roof inside a compound or kept in the house). The owned dogs further have to be separated into puppies (less than 3 months) and the adult dogs. The distinction between puppies and older dogs in the ownerless dog population is not possible with any method presented here. Therefore, we considered the owner population as a whole.

The UAV are able to spot ownerless dogs, owned free-roaming dogs and owned confined visible dogs but not puppies, which are too small in size (Fig 9A). The CR model is designed to count ownerless and owned free-roaming dogs (partially or completely) only and therefore do not consider completely confined dogs (Fig 9B). Owned puppies less than 4 months old were not considered in the CR method because they could not be marked (too small in size for the collar) nor spotted during the transect walks (not visible because they tend to stay at home with the bitch). Unlike the two previous methods, the dog census and human: dog ratio

**Table 5. Percentage of dogs spotted in roads, yards and fields and forest in the UAV pictures and percentage of the study zone covered by each of type of land cover.** The percentage of the study zone covered by each type of land use was calculated using QGIS.

| Land cover | La Romana | | Sabaneta | | Poptún study area | |
|---|---|---|---|---|---|---|
| | Percentage of dogs spotted | Percentage of study zone covered | Percentage of dogs spotted | Percentage of study zone covered | Percentage of dogs spotted | Percentage of study zone covered |
| Yards | 0.91 | 0.31 | 0.64 | 0.68 | 0.69 | 0.83 |
| Roads | 0.00 | 0.02 | 0.36 | 0.05 | 0.31 | 0.17 |
| Fields and forest | 0.09 | 0.16 | 0.00 | 0.28 | 0.00 | 0.00 |

methods focus on all types of owned dogs, including puppies, and do not consider ownerless dogs (Fig 9C).

The human: dog ratio method and the dog census detect the same dog populations, thus the results can be directly compared. In our study, the human: dog ratio estimated a population size close to the census value (97/110 = 88%) in La Romana and underestimated the owned dog population size in Sabaneta (161/289 = 56%).

Taking the census data as a reference, the foot-patrol transect survey detected 71% (78/110) of the dogs in Sabaneta and 90% (259/289) in La Romana. This difference can be explained by the presence of puppies and completely confined dogs detected by the dog census and not the foot-patrol transect survey.

We therefore highlight that when comparing FRDD population size estimates to first carefully evaluate the subpopulation captured by the respective method used.

In this study, we used the used the dog census performed by "One Health Poptún" project as a reference. The participation rate was very high with 99% and 98% households who agreed to participate in La Romana and Sabaneta respectively. Because the census survey was performed nearly one year before the UAV and foot-patrol transects surveys, it is likely that the dog census do not represent the exact number of owned dogs living in La Romana and Sabaneta at the time of the study. This is a limitation in our study. However, no event that would have had a significant impact on the dog population size (i.e. deadly infectious disease outbreak, new regulation concerning FRDD) has been reported by the local team during this period (June 2017 to June 2018). Thus, we would not expect a significant change in the total dog population size.

## Method 1: UAV transect survey

In this study, we were able to demonstrate that detecting FRDD using UAV is possible, although challenging. The advantages of using UAV to estimate the size of FRDD population

**Table 6. Dog population size estimation given by the foot-patrol transect survey, the human: Dog ratio and the total owned dog census collected during the "One Health Poptún" project.**

| *Method* Parameter estimated | La Romana | Sabaneta | Poptún study site |
|---|---|---|---|
| *Foot-patrol transect survey* | | | |
| Mean number of owned FRDD (credibility interval) | 78 (74–82) | 259 (243–277) | 413 (354–483) |
| Mean number of ownerless dogs (credibility interval) | 0.12 (0–0.26) | 0.13 (0–0.29) | 0.21 (0–0.46) |
| Mean total number of FRDD (credibility interval) | 78 (74–82) | 259 (243–277) | 413 (355–484) |
| *Human: dog ratio* | | | |
| Owned dog population size | 97 | 161 | - |
| *Total owned dog census* | | | |
| Owned dog population size | 110 | 289 | - |

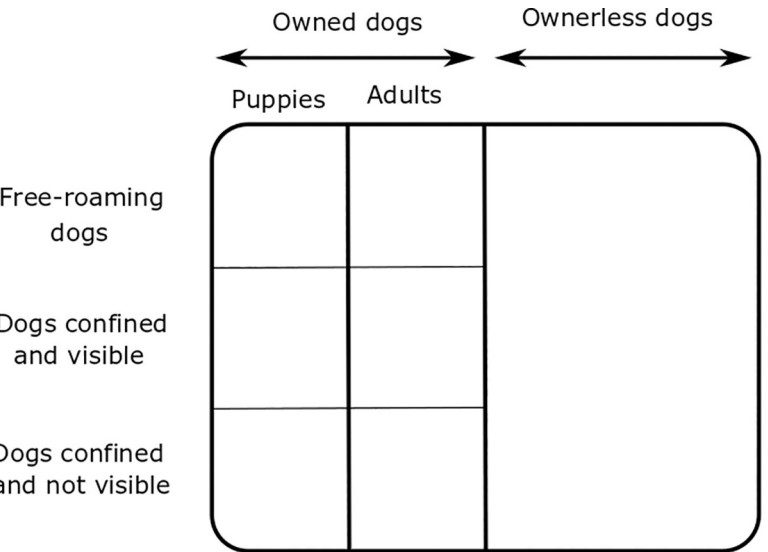

**Fig 8. Structure of the dog population in the study region, Petén department, Guatemala.**

include their capacity of performing flight plan autonomously, in a reproducible way and the possibility to acquire georeferenced data [30]. It has been proven that they could execute high-resolution wildlife aerial survey with low disturbance of the animals and reach areas that are difficulty accessible [30,35]. In our study, the UAV could cover 100% of the one kilometer square area in a mean time of 49 minutes, while the duration of the foot-patrol survey lasted between 58 and 128 minutes and had a lower coverage (Table 3).

Substantially more dogs were observed during the transect walks than on the pictures taken by the UAV. The suggested main reasons for this difference are: a) the quality of the pictures is poor, b), the altitude of the flight is too high to clearly identify the dogs and c) dogs are not visible from the air by the UAV (because they are located in a covered area or because of the distance and angle from the UAV). In the following section, we will develop these three limitations (a-c) and describe potential solutions.

The reasons for poor quality of the pictures were mainly due to the weather conditions. During the UAV's flight, the weather needs to be clear without wind, otherwise it will cause UAV instability, resulting in blurred pictures. It is therefore required to select the days for flying according to the weather conditions.

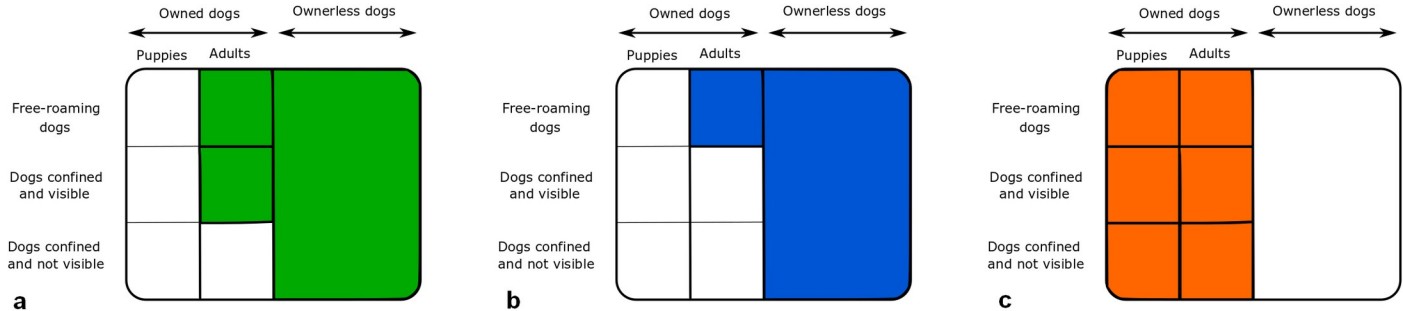

**Fig 9. Study population of the methods used to estimate the size of dog populations.** a. Method using UAV, b. foot-patrol transect survey method and analysis using a CR model, c. dog census and human: dog ratio method. The colorful areas represent the part of the FRDD population covered by each method.

Because of the flight altitude and the small size of a dog, potential dogs could be confused with other animals or items (i.e. rocks) and vice-versa [34]. Doubtful pictures were analyzed by three persons searching for the presence of typical dog shapes. However, if no consensus was found, the observation was not counted as dog, which may have led to dog missed on the pictures. Flying at low altitude is not always possible due to the characteristics of the terrain. In our study, the mountainous landscape in La Romana prevented the UAV from flying below 100m. According to Chrétien et al, wolves could be confused with rocks for flights at 60 meters [34]. Optimal flight altitudes should be identified for the respective species and environment before the use of UAVs. It is therefore recommended to use this method in flat and fully open areas only. An alternative for hilly or mountainous areas is the utilization of flight planning software allowing for terrain following (e.g. UgCS www.ugcs.com or Map Pilot www.mapsmadeeasy.com). These software allow the UAV to fly at a constant level about the ground, regardless of the topography, instead of flying at a constant altitude. Using these type of software, UAV flight could be used in both flat and mountainous landscapes.

The high workload for analyzing pictures to identify the targeted object is one of the drawback of using aerial pictures in animal management and research [30]. To further improve the detection of dogs in the UAV pictures, the utilization of infrared camera and automatic detection of the dogs could be investigated. In the infrared spectrum, it is possible to detect thermal signatures of animals in medium, broken and sparse canopy [44]. Infrared thermal imagery is therefore implementable in more various settings than the visible spectrum. However, the cost of infrared camera is important and be can prohibitive [45]. A cost-benefit analysis would be needed to assess if using infrared imagery would be economically viable in practice. Automatic detection of the dogs using identification and counting algorithms or deep learning could be used to identify and count dogs on the UAV's pictures in both visible and infrared spectrum. Deep learning is part of representation learning's set of methods. In representation learning, machines are fed with raw data and automatically discover the representations needed to detect or classify. Deep learning allows to learn very complex functions by using multiple levels of representation [46]. Algorithms have already been developed for red deer, roe deer, wild boars [33], bison, elks, wolves [34], koalas, kangaroos [30] and cows [47]. Deep learning has been used for automatically identifying, counting and describing wild animals in camera-trap images [48]. Although, the study implemented by Chrétien et al was not completely successful for the identification of wolves in UAV thermal images using a multicriteria object-based image analysis [34], the utilization of artificial intelligence in the study conducted by Gonzalez et al showed good results for small size animals like koalas [30]. It has also been shown in a study involving the identification of animals on 3.2 million camera-trap pictures (Snapshot Serengeti dataset) that deep learning can automatically identify the animal for 99.3% of the pictures with the same accuracy than crowdsourced teams of human volunteers (96.6%) [48]. Thanks to this progress, we are confident that such methods can be developed for dogs. With good quality pictures, dogs could be differentiated from other animals such as calves, goats or sheep in the visible and infrared spectrum. Because deep learning can identify animals with the same accuracy than human beings, manual labor can be saved which will dramatically reduce the cost of analyzing such datasets [48].

Even with optimal conditions, not all FRDD can be detected by UAV (e.g. dogs confined inside a house, dogs lying under trees). Therefore, statistical methods such as used for foot-patrol transects can be used to analyze the dog detection data and obtain reliable estimations of FRDD population size. In this study, we aimed at using a CR method. However, the marking of the dogs was not visible in pictures mainly due to the altitude of the drone flights and the collar being hidden in the neck of the dog. To overcome this limitation, marks clearly visible from above (e.g. red paint on the back of the dogs) should be used. The CR model used to

analyze the foot-patrol survey data performed well but is not adapted to the UAV because of the impossibility to differentiate confined and free-roaming dogs on the UAV pictures. The model takes into account the probability of a dog to be confined and if confined dogs are consider as recaptured (i.e. a dog located in a close yard), the estimation will be biased. Alternatively, spatial capture recapture model or spatial models that do not require recapture could be applied to analyses the UAV dog counts [49–51]. The UAV coordinates (i.e. longitude, latitude and altitude) are recorded for each picture. Software can use this data to create digital spatial model (e.g. Pix4Dmapper, https://www.pix4d.com/product/pix4dmapper-photogrammetry-software). However, to obtain good estimates from population size estimation models, information on how FRDD interact with their environment is crucial. First estimations have clearly shown that we cannot assume equal distribution of the dogs in the different land cover types (Table 5). The detection of dogs in roads are overrepresented, which can result from a higher detection probability in roads than in yards and/or FRDD spending more time on roads than in yards. The researchers who reviewed the pictures reported that dogs were easier to spot on the roads than the yards because yards are often covered by trees and other places where dogs can hide. On the opposite, barely no dogs have been spotted in forest and fields. This was expected because dogs are not visible in this type of land cover, which resulted in the exclusion of those pictures in the first step of the UAV picture analysis (Fig 3). However, it is possible that dogs roaming in the forest were missed and models should account for those dogs. Further investigation on land type use of FRDD is therefore required to inform spatial models.

An important point to take into consideration is the prevalent gray area regarding country regulations when employing UAVs in research related to public health, especially when flying over private and public areas at low altitudes, as it elicits risks for breach of privacy and public safety. In applications such as this one, ethical approval has to be granted by ERBs who might demand to follow compliance with detailed protocols of human subjects' research, even if no direct intervention with people (such as interviews) is foreseen. This can be a cumbersome process if no precedents are established in the country or study area, increasing time and resources needed to comply with varied interpretations of an appropriate ethical standard. However, to ensure a best practice approach in this emerging field of application, it is recommended to be overly cautious in securing local communities' formal approval of the research intervention, being mindful to sociocultural expectations, traditional governance, and consuetudinary law.

## Method 2: Foot-patrol transect survey and CR model

The CR model based on the foot-patrol survey data estimated the size of FRDD populations in a satisfactory way in this study. The presence of puppies and completely confined dogs can explain the difference between the estimate given by this method and the dog census. CR methods already demonstrated efficiency in the literature. It is simple method to provide population size estimates [17] and they are useful to understand population dynamics of FRDD [20].

The prior data on dog keeping practices and owned to ownerless dog ratio were derived from the dog owner's interviews. We decided to interview the dog owners, as also performed in previous studies [15,24,43,52], because non-dog owning resident would probably not have any information about dog keeping practices. In addition, we believed that dog owners might be more aware of the dog population structure and including non-dog owning residents could have biased the result of the owned to ownerless dog ratio. Non-dog owning residents could assume that an owned dog is ownerless because they did not know the dog nor the owner. Dog owners would be more likely to know other dog owners, as well as their dogs. Not all dog-

owning household could be included in the study but the main reason for non-participation was the absence of the dog or the owner. This represents the authors' opinion and might be a limitation of the study. Only very few people refused to participate (the exact proportion has not been recorded). It is unlikely that the refusals would significantly impact our results.

To assess the impact of the prior values (i.e. confinement probability, ownerless to owned dog ratio and number of owned dogs living in the study site) on the three outcomes (i.e. number of owned FRDD, number of ownerless dogs and total number of FRDD), we increased the range of the confinement probability uniform distribution, the expected value of the ownerless to owned dogs ratio (log normal distribution) and the range of the number of owned dog uniform distribution. The maximum value of confinement probability was increased to 0.5 in La Romana and Sabaneta and to 0.75 in Poptún. The mean value of the three outcomes were impacted of maximum 1%. A gamma distribution, allowing a higher ownerless to owned dog ratio than the log normal distribution used previously in the model, was used to assess the impact of this prior. The choice of the prior distribution had an impact on the estimated number of owned FRDD and ownerless dogs but decreased mean total number of FRDD by a maximum of 5.8% (S7 Table). Therefore, this model should not be used to calculate the size of owned dog population in a region where ownerless dogs are present and there are not reliable prior information on ownerless dog population. The increase of the maximum value of owned dog distribution had an impact of less than 1% on the outcomes. This method performed well in counting free-roaming dogs, especially when ownerless dogs are present and prior information is available. However the drawback of not counting puppies remains.

The CR model used in this study takes into account an estimation of the minimum and a maximum transect coverage but not the actual distance between individual dogs and the transect line and hypothesize that the spatial repartition of marked and unmarked dogs is similar. Individual distance between the dogs and the transect lines have an impact on the detection of the dogs and marked and unmarked dogs spatial repartition potentially differs. Recording the distance between each spotted dog and the transect line and including it in the CR model would account for the difference in spatial repartition and may improve the accuracy of the estimated FRDD population size [14,50,53,54]. However, because the number of ownerless dogs is negligible and the reason for non-marking owned dogs were mostly the absence of the dogs or the owner during the marking and interview survey, the marked and unmarked dog spatial repartition might have not differed much in this particular study.

## Method 3: The human: Dog ratio

The human: dog ratio method is currently used by the health authorities in Guatemala to estimate the number of vaccines needed for national rabies vaccination campaigns. While this method produced relatively close results for La Romana, it highly underestimated the population size in Sabaneta, taking the dog census data as a reference. The use of the dog and human census data quantified by "One Health Poptún" project to compute the human: dog ratio results in a value 4.4:1 in La Romana and 2.7:1 in Sabaneta. Although Sabaneta and La Romana are both in a rural setting and located only 30km apart, the human: dog ratio varies strongly. This highlights that an overall country human: dog ratio is not the best way to estimate size of FRDD populations. These findings are consolidated by the results of the dog census performed in Todos Santos Cuchumatán neighbourhoods which results in a human:owned dog ratio varying from 4:1 to 12.3:1 [26] and the findings of the studies currently ongoing at the UVG (unpublished data). It would be interesting to assess if the human: dog ratio is dependent on factors such as socioeconomic or environmental factors. Studies performed in Africa indicated

that religion and the need for guarding property and protecting livestock could be part of those factors [55,56].

## Conclusion

This is the first study published on the investigation of UAV utilization to collect information for free-roaming domestic dogs population size estimation. We showed that UAV could be used to detect dogs. However, the quality of the pictures has to be improved to obtain more reliable counts and spatial models should be used to accurately estimate the size of the dog population. The utilization of infrared thermal imagery and automatic detection of the dogs could also be investigated. The foot-patrol transect survey method estimated the FRDD population size in a satisfactory way. The last method, based on the dog: human ratio tended to underestimate the size of the FRDD population in our study areas. This study may be useful to give directions on how to design efficient application of UAV for this purpose. It also highlights that the aim for the dog populations size estimation influence the method to be selected.

## Supporting information

**S1 Fig. Transects lines in the three study areas, Petén, Guatemala.** A. La Romana, B. Sabaneta, C. Poptún study area. The green and the red dots are the starting and ending points, respectively.
(PDF)

**S2 Fig. Ratio of estimated number of ownerless to owned dogs in Poptún, Guatemala.**
(PNG)

**S1 Table. GPS coordinates of the three study sites.**
(PDF)

**S2 Table. Questionnaire survey used to collect information about confinement practices and proportion of dogs based on their ownership status in the three study sites, Guatemala.** The interviewees were dog owners whose household is located in the three study sites.
(PDF)

**S3 Table. Dog detection table of the UAV transect survey.**
(XLSX)

**S4 Table. Data collection during the dog owner's interview using a questionnaire.**
(CSV)

**S5 Table. Confinement practices of dog's owners in Petén department, Guatemala.**
(PDF)

**S6 Table. Number of marked and unmarked dogs recaptured during the ground transects.**
(PDF)

**S7 Table. Impact of the prior ownerless to owned dogs ratio distribution on the model outcomes.**
(PDF)

**S1 File. Script of the capture-recapture model implemented on OpenBUGS.**
(PDF)

## Acknowledgments

We thank Pablo Ax, Claudia Caal, Pablo Chub, Julio Eufragio, Dione Méndez, Milton Oliva, Jorge Paniagua, German Rodas, and Alexis Tut for their help and excellent work on the field; Wendy Hernández, Ramón Medrano and Adan Real for reviewing the drone pictures; Sonja Hartnack and Beatriz Vidondo for their feedback on the capture-recapture model; Jakob Zins-stag for giving us the opportunity to use the "One Health Poptún" project setting to implement this study and all the dog owner's for their participation.

## Author Contributions

**Conceptualization:** Charlotte Warembourg, Monica Berger-González, Danilo Alvarez, Salome Dürr.

**Data curation:** Charlotte Warembourg, Monica Berger-González, Danilo Alvarez, Filipe Maximiano Sousa.

**Formal analysis:** Charlotte Warembourg.

**Funding acquisition:** Monica Berger-González, Filipe Maximiano Sousa, Salome Dürr.

**Investigation:** Charlotte Warembourg, Filipe Maximiano Sousa, Alexis López Hernández, Pablo Roquel.

**Methodology:** Charlotte Warembourg, Monica Berger-González, Danilo Alvarez, Joe Eyerman, Merlin Benner, Salome Dürr.

**Project administration:** Monica Berger-González, Danilo Alvarez, Salome Dürr.

**Resources:** Monica Berger-González, Danilo Alvarez, Joe Eyerman, Merlin Benner, Salome Dürr.

**Supervision:** Monica Berger-González, Danilo Alvarez, Salome Dürr.

**Validation:** Charlotte Warembourg, Salome Dürr.

**Visualization:** Charlotte Warembourg.

**Writing – original draft:** Charlotte Warembourg, Monica Berger-González, Danilo Alvarez.

**Writing – review & editing:** Charlotte Warembourg, Monica Berger-González, Danilo Alvarez, Filipe Maximiano Sousa, Joe Eyerman, Merlin Benner, Salome Dürr.

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
