## [Decision Letter · Decision Letter 0]

2 Dec 2019

PONE-D-19-29661

Using Unmanned Aerial Vehicles (UAV) for free-roaming dog population size estimation

PLOS ONE

Dear Charlotte Warembourg 

Thank you for submitting your manuscript to PLOS ONE. After careful consideration, we feel that it has merit but does not fully meet PLOS ONE’s publication criteria as it currently stands. Therefore, we invite you to submit a revised version of the manuscript that addresses the points raised during the review process.

Many thanks for submitting your manuscript to PLOS One

Your manuscript was sent to three people to review, and they have suggested some revisions

They have provided some very good helpful comments aimed at improving the manuscript

If you can write a detailed response to reviewers, as the manuscript will be returned back to the same reviewers after resubmission if possible.

I wish you the best of luck with your revisions

Many thanks

Simon

We would appreciate receiving your revised manuscript by Jan 16 2020 11:59PM. To enhance the reproducibility of your results, we recommend that if applicable you deposit your laboratory protocols in protocols.io, where a protocol can be assigned its own identifier (DOI) such that it can be cited independently in the future. For instructions see: http://journals.plos.org/plosone/s/submission-guidelines#loc-laboratory-protocols

We look forward to receiving your revised manuscript.

Kind regards,

Simon Russell Clegg, PhD

Academic Editor

PLOS ONE

Journal Requirements:

2. In your Methods section, please provide additional location information of the study areas, including geographic coordinates for the data set if available.

4. Thank you for stating the following in the Financial Disclosure section:

Funding for this research was provided by the Albert-Heim-Stiftung Project Nr. 132 – (http://www.albert-heim-stiftung.ch, awarded to SD), the Spezialisierungskommission of Bern University (SpezKo) (https://www.vetsuisse.unibe.ch/weiterbildung/spezialisierungskommission/index_ger.html, awarded to FMS), the Wolfermann-Nägeli-Stiftung Nr. 2018/28 (awarded to SD) and the Swiss Programme for Research on Global Issues for Development (r4d programme), Project Nr. 160919, (http://p3.snf.ch/project-160919). The two UAV used during the study were provided by RTI International and the Universidad del Valle.

We note that one or more of the authors are employed by a commercial company: RTI International and/or Remote Intelligence, LLC,

Reviewers' comments:

Reviewer's Responses to Questions

**Comments to the Author**

1. Is the manuscript technically sound, and do the data support the conclusions?

Reviewer #1: Yes

Reviewer #2: Partly

Reviewer #3: Yes

2. Has the statistical analysis been performed appropriately and rigorously? 

Reviewer #1: Yes

Reviewer #2: No

Reviewer #3: No

3. Have the authors made all data underlying the findings in their manuscript fully available?

Reviewer #1: Yes

Reviewer #2: No

Reviewer #3: No

4. Is the manuscript presented in an intelligible fashion and written in standard English?

Reviewer #1: Yes

Reviewer #2: No

Reviewer #3: Yes

5. Review Comments to the Author

Reviewer #1: At the outset, I would congratulate the authors for choosing a very challenging subject and appreciate the rigorous field work untaken.

The manuscript compares the population size estimates of dogs obtained through three methods, (a) a direct count through aerial photographic count from four survey sessions (b) Bayesian statistical model using data from capture –recapture techniques and priors from household interviews, and (c) through human: dog population ratios estimation through information collected through interviews of selected sample of residents, with the owned dog population estimate obtained through census.

While the procedure of the methods used have been described well, the very purpose of designing such a study needs to be brought out emphatically, which I feel is lacking as the manuscript stands now. In other words why would one attempt UAV aerial counting? Are the methods previously used are inadequate? If yes, how will this method mitigate? I would suggest also discussing the conomics of using such methodology over other methods.

Also, the title of the manuscript includes only one aspect (i.e. use of UAV) and precludes other methods used or the important aspect that the manuscript compares the different methods.

The detailed comments are as follows:

Line 73-75: Although, underestimation could be an issue with some of the methods, it is not understood how capture-recapture technique precludes the ownerless free-roaming dogs. It will be good to elaborate and also cite the work where this is the case.

Line 77-78: It will be good if you could provide examples of priors that have been used earlier with references, please.

Lines 106-110: The objective of the study aligns with the title, however, when we look at the results a basic query that comes up id that if using UAV is not helpful and if we still have to compare it to census (I am not sure if you consider it the gold standard), why use it at all! It is in this context that I suggest that the Bayesian method and human : dog ratio method should be included in the aim as well (and also in the title).

Line 129: Please provide the date the data was accessed

Line 145: Please provide the source.

Line147-151: How can one differentiate between free-roaming and confined dog from the aerial pictures (also the how can one differentiate between owned confined and owned free-roaming dog from the serial pictures)? Suggest provide an explanation here.

Line 202-203: Why was morning and late afternoon time chosen?

Table 2: Why does the flight time vary considerably? There is a wide time difference between the flight times within similar temporal sessions. However, the difference in flight times is not associated with the number of pictures taken. Can you elaborate on this?

Line 233-240: Suggest provide a flow chart for easy understanding.

Line 272: ‘Cultural beliefs’ such as?

Line 273: How did you know a dog was four months? (please mention it was based on visual appreciation, if it was so).

Line 277-286: Were the interviews conducted on all dog owning households? Or on selected ones? If on selected ones, how was the sampling done? Why were the non-dog owning residents not included? If only dog owners were included would it not introduce a bias in the ’prior’ estimation?

Line 294: What about the dogs in the buffer zone?

Line 309: Do you mean ‘lose’?

Line 314: What is the basis of 10% assumption? (In other words what is the likely dog population size for this assumption of 10%)?

Line 398-400: Probably better suited to be part of discussion section.

Lines 401-403; 406-408 & Table 5: Suggest round-off the number of dogs to whole numbers.

Table 2, 4 & 5: please give the first row/column heading.

Line 436: Did you mean the owned pups? In that case what about the ownerless pups?

Line: Please say how the dog census was done, and when.

Line 448- 452: Please reword to provide clarity.

Line 450: Which two methods? Please explain for the benefit of the reader.

Line 476-477: “fully open environment”. Is it practically possible to drive all free-roaming dogs into an open environment to count them? I am not sure about it.

Secondly, I think you already used the temporal settings as the ones you are suggesting (morning and late afternoon), but it yielded low counts!

Line 486: Suggest explain briefly about ‘deep learning’ the references are great, but a brief context here will help the reader).

Line 481: Infrared thermal imagery methods: how would they preclude other animals in rural areas such as calfs/foals/sheep/goat etc.? Also will it be economically viable?

Line 506: If wolves can be confused for rocks, the vice versa could also be true, and we can miss some dogs presuming they are rocks!

Please also discuss the advantages of the use of UAV in dog population estimation supported by refernces.

Reviewer #2: The authors used UAVs, foot-patrol transect surveys, and human:dog ratios to attempt to estimate free-roaming dog population sizes for three separate study areas in Guatemala. Although I agree that the scientific foundation for attempting to make such comparisons is generally solid, I have a few major criticisms of the authors’ approach to arriving at those comparisons, based entirely on the survey designs and statistical analyses.

1. The data collected from the UAV surveys were not analyzed. The authors treat these data solely as raw counts of the number of dogs visible in the photographs obtained via the UAVs. At best, these results are simply indices, not estimates, of the minimum population sizes of dogs that are detectable via aerial surveys. As the authors note on lines 233-240, 384-388, and 464-468, these UAV data are imperfect detection data with a large proportion of dogs being undetected. The reasons for this are numerous, some of which the authors acknowledged (poor quality photos, vegetation cover, flight altitude, etc.), whereas others were not considered (e.g., distance and angle of dog from UAV transect when photo was taken). Yet, none of those factors were included in any formal analytical framework. To be clear, the UAV transect surveys are analogous to airplane or helicopter transect surveys, which constitute distance sampling with imperfect detection. Thus, the authors need to analyze the UAV data collected from Sabaneta and Poptun using a model that accounts for the distances between the UAV transect and the locations of detected dogs, as well as imperfect detection. Given many of the dogs in the study area were marked and had known identity, these data represent mark-resight distance sampling data, which the modified MRDS models developed by Borchers et al. (2015; J. Am. Stat. Assoc. 110:195-204) are well suited to analyzing. Alternatively, because the spatial locations of each UAV transect and detected dog are available, the authors could discretize the survey area as a gridded or transect state space (Russell et al. [2012] J. Wildlife Manag. 76:1551-1561; Thompson et al. [2012] J. Wildlife Manag. 76:863-871; Morin et al. [2016] 80:824-836; Murphy et al. [2018] Ecosphere 9:e02479) and apply a spatially explicit mark-resight model to estimate population size (Chandler and Royle [2013] Annals Appl. Stat. 7:936-954; Sollmann et al. [2013] Ecology 94:553-559; Murphy et al. [2019] Sci. Reports 9:4590).

2. The foot-patrol transect surveys also represent distance sampling with imperfect detection. However, the authors’ model does not accommodate any distance measurements and therefore heterogeneity in detection due to spatial proximity between the transects and dogs is unmodeled, leading to estimates that are likely biased to an unknown magnitude. These data are also mark-resight distance sampling, given detections of both marked and unmarked dogs were recorded at varying distances from the walked transects. Thus, similar to above for UAV transects, an MRDS or spatial mark-resight model should be used to analyze these detection data.

3. “Bayesian statistical model” is an extraordinarily vague term that does not accurately represent the model that the authors used to analyze the foot-patrol transect survey data. The authors must be specific about this model throughout the manuscript: This is a non-spatial mark-resight model, with some minor modifications for mark loss and ratio of owned:ownerless dogs, which was fit in a Bayesian inferential framework. Mark-resight models that accommodate mark loss have been available in the frequentist inferential framework for a decade (e.g., McClintock et al. [2009] Biometrics 65:237-246; McClintock [2018] Chapter 18 in Cooch & White (eds.), Program MARK: A Gentle Introduction). Thus, the fact that the authors fit their model in a Bayesian framework is relatively unimportant. Furthermore, as noted above, the specification of the authors’ model was insufficient for accurately estimating dog population size from the foot-patrol transect data. At minimum, a model needs to be used that accounts for spatial distances between the transects and detected dogs; evaluates variation in detection probability due to survey, environmental, and/or individual covariates; and does not assume that both marked and unmarked dogs have the same spatial distributions relative to the marking/transect locations.

Minor Comments:

Data: The detection data derived from the photographs, not the photographs themselves, need to be provided. The results are not reproducible without the detection data.

Lines 4-6: This title is misleading. It suggests that UAVs were used to estimate dog population size, but they were not. The UAV surveys only provide a count of minimum number of dogs, because the data were not analyzed using any modeling approach.

Lines 73-75: These statements are not true. Capture-recapture and mark-resight models that account for such heterogeneity via either finite mixtures (Pledger [2000] Biometrics 56:434-442; Gardner et al. [2010] J. Wildlife Manag. 74:318-325) or individual covariates (Huggins [1989] Biometrika 76:133-140) have been around for decades.

Lines 76-78: As noted above, the term “Bayesian statistical models” is too vague for use here. Bayesian is just an inferential framework and all models are statistical models. Additionally, as noted above, frequentist models are also able to account for heterogeneity in the population.

Line 356: Again, this is a capture-recapture (mark-resight) model that was fit in a Bayesian framework, NOT a “Bayesian statistical model based on capture-recapture data”.

Lines 264-265: Why were the location data recorded by these GPS collars not incorporated in a spatially explicit model? Doing so would improve accuracy of the population size estimates (Sollmann et al. 2013, Murphy et al. 2019).

Lines 292-300: This describes mark-resight distance sampling data, but the authors’ model is not a mark-resight distance sampling model.

Lines 384-388: Then why was heterogeneity in detection due to these factors not included in a model to formally analyze the UAV data?

Lines 419-420: Misleading statement. The UAV data were not used to estimate population size, only to produce a minimum count.

Lines 464-474: Then a model needs to be used that incorporates heterogeneity in detection.

Lines 493-498: Why was this not done? This is a major weakness of the paper and study.

Lines 533-544: Capture-recapture models fit in a frequentist framework can also do this, so this statement seems irrelevant.

Reviewer #3: I revised the manuscript "Using Unmanned Aerial Vehicles (UAV) for free-roaming dog population size estimation" by Warembourg et al.

The topic of this study is of interest and the results are novel and useful. The unprecedented use of drones to estimate dog abundance is relevant.

On the other hand, some points need to be properly adressed to improve the quality of the manuscript.

-The abstract must present the values of population size estimates. This will be useful for a better understanding of the poor sensitivity of UAV.

-I believe that the inclusion of pictures and/or photos of study areas would be important for a better understanding of the characteristics of the regions, especially to an international readership.

-The"One Health Poptun" research and methods of domestic animal census should be better presented.

-The small number of flights should also be pointed as a factor that limited the sensitivity of UAV. Authors must justify why they have only conducted 11 flights.

-I understand the ethical issues that prevent authors from providing all drone images. However, some pictures should be inserted in the article to exemplify information presented. This is possible and does not violate ethical standards.

Suggestions.

-"Pictures that could not be analysed due to poor quality"

-Pictures in which dogs could be identified.

-Pictures that generated doubts.

-Pictures mentioned in lines 243-245 ("doubtful observations resembling dogs but not presenting dog characteristics, such as ears, thorax, pelvis, limbs or tails, were not counted as dogs").

- Pictures demonstrating the effect of climatic conditions on identification.

-Photographs made during transect walks.

-Concernig the information presented in lines "243-245": "doubtful observations resembling dogs but not presenting dog characteristics, such as ears, thorax, pelvis, limbs or tails, were not counted as dogs".

Why did the authors adopt this criterion?

What are the implications for the sensitivity of the technique?

It is highly recommended that further analysis with an alternative criterion be conducted by the authors.

-The effects of other study limitations should be further discussed:

Non-collared dogs (lines 269-271)

Refusals (lines 271-273).

-With regard to the figures

-Figure 3 should be better explained in the caption.

-I did not understand the figure 4.

-I believe that figure 5 should be inserted in the methodology.

-Authors should check all links, as some pages did not open.

6. PLOS authors have the option to publish the peer review history of their article (what does this mean?). If published, this will include your full peer review and any attached files.

Reviewer #2: No

Reviewer #3: No

---

## [Author Response · Author response to Decision Letter 0]

6 Feb 2020

We thank the reviewers for the highly relevant comments. We addressed them in details in the response to the reviewers document.

---

## [Decision Letter · Decision Letter 1]

25 Feb 2020

PONE-D-19-29661R1

Estimation of free-roaming domestic dog population size: investigation of three methods including an Unmanned Aerial Vehicle (UAV) based approach

PLOS ONE

Dear Dr Warembourg

Thank you for submitting your manuscript to PLOS ONE. After careful consideration, we feel that it has merit but does not fully meet PLOS ONE’s publication criteria as it currently stands. Therefore, we invite you to submit a revised version of the manuscript that addresses the points raised during the review process.

Many thanks for resubmitting your manuscript to PLOS One

One reviewer has made some very minor comments. If you could address those, with a brief response to reviewers, then I can recommend the manuscript for publication

I wish you the best of luck with your minor revisions

Many thanks

Simon

We would appreciate receiving your revised manuscript by Apr 10 2020 11:59PM. To enhance the reproducibility of your results, we recommend that if applicable you deposit your laboratory protocols in protocols.io, where a protocol can be assigned its own identifier (DOI) such that it can be cited independently in the future. For instructions see: http://journals.plos.org/plosone/s/submission-guidelines#loc-laboratory-protocols

We look forward to receiving your revised manuscript.

Kind regards,

Simon Russell Clegg, PhD

Academic Editor

PLOS ONE

Reviewers' comments:

Reviewer's Responses to Questions

**Comments to the Author**

1. If the authors have adequately addressed your comments raised in a previous round of review and you feel that this manuscript is now acceptable for publication, you may indicate that here to bypass the “Comments to the Author” section, enter your conflict of interest statement in the “Confidential to Editor” section, and submit your "Accept" recommendation.

Reviewer #1: All comments have been addressed

Reviewer #2: (No Response)

2. Is the manuscript technically sound, and do the data support the conclusions?

Reviewer #1: Yes

Reviewer #2: Partly

3. Has the statistical analysis been performed appropriately and rigorously? 

Reviewer #1: Yes

Reviewer #2: No

4. Have the authors made all data underlying the findings in their manuscript fully available?

Reviewer #1: Yes

Reviewer #2: Yes

5. Is the manuscript presented in an intelligible fashion and written in standard English?

Reviewer #1: Yes

Reviewer #2: Yes

6. Review Comments to the Author

Reviewer #1: The article will immensely contribute towards future research on dog population management. Recommended for acceptance.

Reviewer #2: I applaud the authors for making quality revisions to their manuscript and addressing our previous comments. As a result of their efforts, I think the manuscript is very much improved. I do have some lingering minor comments for the authors to consider:

1) Lines 226-227: I am still concerned about the choice to not use the GPS locations from the marked dogs, whose collars had a GPS chip, in the analyses. The locations obtained from those GPS chips could help the authors determine which dogs in the photos from UAV survey were marked versus unmarked. With 71 and 96 dog detections via the UAV flights at Poptun and Sabaneta, respectively, sufficient data exist to estimate the population size for both of those locales using a mark-resight model if the locations from marked dogs' GPS collars are used to differentiate between dogs in photos by their marked vs unmarked status. A similar approach is used when sampling via camera-traps and some animals in a population have GPS collars (e.g., see Whittington et al. [2018]: https://besjournals.onlinelibrary.wiley.com/doi/full/10.1111/1365-2664.12954; Murphy et al. [2019]: https://www.nature.com/articles/s41598-019-40926-7, among others). I see no reason why this cannot be applied to the UAV data to discriminate between marked and unmarked dogs, which would allow the authors to produce population size estimates from UAV flights for two of the three sampled locales.

2) Table 6: The caption indicates that estimates from the UAV survey are in included, but they are not. Also, why are the values for mean number of ownerless dogs all zeros?

3) Lines 512-519: This information also needs to be presented upfront in the Methods section for each approach. Also, if the CR model does not "consider confined dogs" (line 514), then why is a confinement probability included in the model?

4) Lines 577-578: This is a very questionable recommendation. If researchers follow this, then considerable sampling bias will be induced in the estimates.

5) Lines 592-608: Way too much discussion and space devoted to something that the authors did not use. This stuff about deep learning needs to be reduced to 2-3 sentences maximum.

6) Line 610: But earlier in the Methods and Discussion, the authors stated that confined dogs can be detected by UAVs. Which is it?

7) Lines 214-215: Deploying GPS collars on dogs would probably be a more fruitful approach, because the locations from each marked dog can be used to determine where and when a marked dog was located relative to the UAV flight time and flight path.

8) Lines 623-634: All of these are not restricted to spatial models, but apply to non-spatial models as well, including the authors' model.

9) Lines 658-662: These are very strong assumptions that may not represent reality.

10) Lines 675-679: Unclear why this would not be critical if the aim is to estimate FRDD population size, given a 5.8% influence is not negligible.

11) Lines 686-692: The authors need to provide relevant citations for these statements.

7. PLOS authors have the option to publish the peer review history of their article (what does this mean?). If published, this will include your full peer review and any attached files.

Reviewer #2: No

---

## [Author Response · Author response to Decision Letter 1]

11 Mar 2020

Many thanks for the editor's and reviewers' work. We addressed the comments in detail in the response to the reviewers.

---

## [Editor Report · Decision Letter 2]

17 Mar 2020

Estimation of free-roaming domestic dog population size: investigation of three methods including an Unmanned Aerial Vehicle (UAV) based approach

PONE-D-19-29661R2

Dear Dr. Warembourg

We are pleased to inform you that your manuscript has been judged scientifically suitable for publication and will be formally accepted for publication once it complies with all outstanding technical requirements.

With kind regards,

Simon Russell Clegg, PhD

Academic Editor

PLOS ONE

Additional Editor Comments (optional):

Many thanks for resubmitting the manuscript to PLOS One, and for your response to reviewers comments

I have reviewed the manuscript and recommended it for acceptance

You should hopefully hear from the editorial office soon

It was a pleasure working with you and I wish you all the best for your future research

Many thanks

Simon
---

## [Editor Report · Acceptance letter]

25 Mar 2020

PONE-D-19-29661R2 

Estimation of free-roaming domestic dog population size: investigation of three methods including an Unmanned Aerial Vehicle (UAV) based approach 

Dear Dr. Warembourg:

I am pleased to inform you that your manuscript has been deemed suitable for publication in PLOS ONE. Congratulations! Your manuscript is now with our production department. 

With kind regards,

on behalf of

Dr. Simon Russell Clegg 

Academic Editor

PLOS ONE